# Berberine Reduces Lipid Accumulation in Obesity via Mediating Transcriptional Function of PPARδ

**DOI:** 10.3390/ijms241411600

**Published:** 2023-07-18

**Authors:** Jia-Wen Shou, Pang-Chui Shaw

**Affiliations:** 1Li Dak Sum Yip Yio Chin R&D Centre for Chinese Medicine, The Chinese University of Hong Kong, Hong Kong, China; jiawenshou@cuhk.edu.hk; 2School of Life Sciences, The Chinese University of Hong Kong, Hong Kong, China; 3State Key Laboratory of Research on Bioactivities and Clinical Applications of Medicinal Plants and Institute of Chinese Medicine, The Chinese University of Hong Kong, Hong Kong, China

**Keywords:** berberine, obesity, PPARδ, adipogenesis

## Abstract

Obesity is defined as a dampness-heat syndrome in traditional Chinese medicine. Coptidis Rhizoma is an herb used to clear heat and eliminate dampness in obesity and its complications. Berberine (BBR), the main active compound in Coptidis Rhizoma, shows anti-obesity effects. Peroxisome proliferator-activated receptors (PPARs) are a group of nuclear receptor proteins that regulate the expression of genes involved in energy metabolism, lipid metabolism, inflammation, and adipogenesis. However, whether PPARs are involved in the anti-obesity effect of BBR remains unclear. As such, the aim of this study was to elucidate the role of PPARs in BBR treatment on obesity and the underlying molecular mechanisms. Our data showed that BBR produced a dose-dependent regulation of the levels of PPARγ and PPARδ but not PPARα. The results of gene silencing and specific antagonist treatment demonstrated that PPARδ is key to the effect of BBR. In 3T3L1 preadipocytes, BBR reduced lipid accumulation; in high-fat-diet (HFD)-induced obese mice, BBR reduced weight gain and white adipose tissue mass and corrected the disturbed biochemical parameters, including lipid levels and inflammatory and oxidative markers. Both the in vitro and in vivo efficacies of BBR were reversed by the presence of a specific antagonist of PPARδ. The results of a mechanistic study revealed that BBR could activate PPARδ in both 3T3L1 cells and HFD mice, as evidenced by the significant upregulation of PPARδ endogenous downstream genes. After activating by BBR, the transcriptional functions of PPARδ were invoked, exhibiting negative regulation of CCAAT/enhancer-binding protein α (Cebpα) and Pparγ promoters and positive mediation of heme oxygenase-1 (Ho-1) promoter. In summary, this is the first report of a novel anti-obesity mechanism of BBR, which was achieved through the PPARδ-dependent reduction in lipid accumulation.

## 1. Introduction

Obesity is defined as an abnormal or excessive fat accumulation, according to the World Health Organization, which is posing risks worldwide to people of all ages, genders, and socioeconomic statuses. The World Obesity Federation estimated that one billion people, including one in five women and one in seven men, will be living with obesity by 2030 [1]. Obesity is closely related to numerous health problems, such as cardiovascular disease, diabetes, and cancer [2]. Additionally, the COVID-19 outbreak led to disrupted routine lifestyles, increased stress, and less opportunity for physical activity, so many people experienced weight gain [3]. A large body of clinical evidence suggests that people with obesity are the most vulnerable to morbidity and even mortality owing to COVID-19, as they are more prone to developing severe COVID-19 pneumonia [4].

Although no cure currently exists for obesity, current therapeutic measures for obesity management mainly include lifestyle interventions, medication, and surgery [5]. Lifestyle interventions such as diet and exercise are the first line of defense in obesity management and can effectively reduce weight and improve health outcomes [6]. However, epidemic and clinical surveys have indicated that those with obesity are reluctant to adjust their lifestyle or do not do so consistently to maintain long-term modification [7]. Therefore, medication might be an alternative method to achieve weight loss. To date, several oral weight loss drugs have been approved by the FDA, such as orlistat, naltrexone, lorcaserin, and phentermine, which are effective but still produce many adverse effects, namely cardiovascular events, suicidality, abuse risk [8,9], and cancer [10]. Therefore, bariatric surgery, including gastric bypass and sleeve gastrectomy, has become a popular option for those with severe obesity, but the procedure also carries risks and complications [11]. Mounting evidence shows that this surgery might negatively affect bone health [12] and lead to dumping syndrome [13]. Consequently, the development of safer and more effective medications has emerged as a necessity.

Peroxisome-proliferator-activated receptors (PPARs) refer to a family of ligand-inducible transcription factors that regulate essential metabolic activities and represent important targets for metabolic syndrome [14]. PPARs are promising candidates to treat obesity [15,16]; however, no commercial PPAR-dependent anti-obesity agents have been developed.

In terms of traditional Chinese medicine, those with obesity have internal heat and phlegm accumulation, qi (vital energy) stagnation, exogenous evil invasion, and toxin excretion blockage, so obesity is classified as a dampness-heat syndrome [17]. Heat-clearing herbs have been traditionally used to treat obesity, of which Coptidis Rhizoma is a good example [17]. Coptidis Rhizoma is classified as a xie xia herb (laxative) in Shennong’s Classic of Materia Medica and has been widely used to clear heat and eliminate dampness in obesity and its complications in ancient China [18,19,20]. Coptidis Rhizoma and its main active compound, berberine (BBR), are beneficial in treating obesity [18,21]. BBR has been reported to show good anti-obesity effects through suppressing appetite [22], activating thermogenesis [23], inhibiting adipogenesis [24], and regulating gut microbiome [21]. Though mounting evidence suggests that BBR is a promising modulator of PPARs [25,26,27], whether PPARs are involved in the anti-obesity effect of BBR remains unrevealed.

In this study, we examined the role of PPARs in the effect of BBR against 3T3L1 preadipocytes and HFD-induced obesity in mice. Lipid accumulation, mouse weight, and biochemical parameters were determined to evaluate the efficacy of BBR. Additionally, gene silencing and specific antagonist treatment were used to verify the role of PPARs. The mechanism through which BBR affects PPARs was studied using qPCR, immunoblotting, immunostaining, and chromatin immunoprecipitation (ChIP)-PCR/qPCR analysis.

## 2. Results

### 2.1. BBR Reduced Lipid Accumulation in 3T3L1 Cells

The structure of BBR is shown in Figure 1A. A mixture of 3-isobutyl-1-methylxanthine, dexamethasone, and insulin (MDI) was used to induce 3T3L1 preadipocytes to differentiate into mature adipocytes (Figure 1B). Nontoxic doses of BBR (Appendix A) were added to 3T3L1 cells on day 0 (pre-MDI), day 2 (during MDI), or day 4 (post-MDI) for 48 h of treatment. As shown in Figure 1C, BBR dose-dependently inhibited lipid accumulation in 3T3L1 cells at all three treatment time points. Of note, BBR showed the most potent inhibition of lipid accumulation when added during MDI induction, producing a reduction of 40% lipid content when compared with the control group (Figure 1C,D).

### 2.2. PPARδ Was Involved in the Effect of BBR on 3T3L1 Cells

We studied the role of PPARs in the effect of BBR on 3T3L1 cells. First, we analyzed the mRNA levels of *Ppars* in BBR-treated 3T3L1 cells: BBR was added during the MDI treatment. We found that BBR dose-dependently increased *Pparδ* level but decreased *Pparγ* expression, while the *Pparα* level remained unchanged (Figure 2A). Next, we detected the expression of PPARδ and PPARγ at the translational level and obtained similar data as that from qPCR (Figure 2B,C).

Then, gene silencing using siRNA transfection was applied to verify the role of PPARs. siRNA transfection was performed on day 0 (Figure 1B), and successful gene silencing was verified with Western blot (WB) (Figure 2D and Appendix A). Neither *Pparα* siRNA nor *Pparγ* siRNA could weaken the effect of BBR, as a reduction in lipid content in both *Pparα* and *Pparγ* siRNA-transfected 3T3L1 cells was not found (Figure 2E). However, silencing PPARδ abolished the effect of BBR to lessen lipid accumulation, as shown in Figure 2E,F. Additionally, a specific antagonist of PPARδ-GSK0660 (GSK) was used to verify the role of PPARδ. At nontoxic doses (Appendix A), GSK was able to antagonize the effects of BBR on 3T3L1 cell lipid accumulation (Figure 2G,H). The ability of BBR to promote PPARδ expression was also weakened (Figure 2I).

Collectively, we concluded that PPARδ is involved in the inhibitory effect of BBR on lipid accumulation.

### 2.3. In Vivo Anti-Obesity Effects of BBR Depended on PPARδ

HFD was administered to mice for six weeks to induce obesity. Firstly, we have performed a preliminary study to confirm the rationale of HFD and BBR doses (Appendix A). Based on the pilot data, we continued to study the in vivo effect of BBR on HFD mice. Compared with the mice administered a normal diet (ND), mice that were heavier (Figure 3A) with higher plasma levels of total cholesterol, glucose, and triglycerides (Appendix A) were considered obese. Three weeks of BBR treatment produced a dose-dependent reduction in mouse weight gain, whereas the presence of GSK, a specific PPARδ inhibitor, weakened the effect of BBR (Figure 3A). Next, we compared the mass of the fat tissues and liver among the different treatment groups. Neither liver nor brown adipose tissue (BAT) mass was altered with BBR treatment in HFD mice, whereas BBR significantly lowered the mass of subcutaneous white adipose tissue (SWAT) and epididymal white adipose tissue (EWAT), which was reversed by GSK treatment (Figure 3B–E and Appendix A).

The hematoxylin and eosin (H&E) staining of the samples demonstrated that HFD administration resulted in larger fat cell size and lipid vacuoles among the liver cells (Figure 4). BBR treatment notably reduced the fat cell size of SWAT (Figure 4A), EWAT (Figure 4B), and BAT (Figure 4D) and lessened the lipid vacuoles inside the liver (Figure 4C and Appendix A). However, the inhibition of PPARδ by GSK reduced the efficacy of BBR (Figure 4). Notably, BBR treatment did not affect the appetite of the mice (Appendix A).

We detected several lipid-related parameters in the plasma, including fasting blood glucose, total cholesterol, high-density lipoprotein cholesterol (HDL), and triglycerides. As obesity is always associated with chronic inflammation and oxidative stress, we also detected the levels of tumor necrosis factor-alpha (TNFα) and malondialdehyde (MDA) in the plasma. BBR was dose-dependently beneficial to correct the biochemical parameters disturbed by HFD administration, whereas GSK mitigated all the effects triggered by BBR (Figure 5).

Together, the results showed that BBR was able to reduce mouse weight gain and fat tissue mass and correct the disturbed biochemical parameters, which were achieved by PPARδ.

### 2.4. BBR Activated PPARδ to Regulate Downstream Genes

We previously found that BBR functions as a potent ligand for PPARδ and subsequently triggers the transcriptional regulation of the downstream genes to exert biological effects [26]. We thereby speculated whether BBR could activate PPARδ in 3T3L1 cells and the EWAT of HFD mice. Two endogenous PPARδ downstream genes, mitochondrial uncoupling protein 2 (*Ucp2*) and pyruvate dehydrogenase kinase 4 (*Pdk4*), were detected via qPCR, and the data showed that BBR could promote the expressions of these two genes, but GSK reduced this effect (Figure 6A). Therefore, we noted that PPARδ was activated both in 3T3L1 cells and HFD mice.

Next, we screened the downstream lipid-accumulation-related genes that might be regulated by BBR-activated PPARδ via qPCR analysis. From a literature review, we found that *Cebpα*, *Cebpβ*, *Pparγ*, *Ho-1*, GATA-binding factor 2 (*Gata2*), *Gata3*, Galectin-3 (*Gal-3*), and fatty acid-binding protein *4* (*aP2*) might be related to the BBR-induced inhibition of lipid accumulation [24,28,29,30,31]. Except for *Gal-3* and *aP2*, we found that *Cebpα*, *Cebpβ*, *Pparγ*, *Ho-1*, *Gata2*, and *Gata3* were regulated by the BBR treatment (Figure 6B and Appendix A); however, only the levels *Cebpα*, *Pparγ*, and *Ho-1* were reversed when PPARδ was inhibited (Figure 6B). Additionally, we detected the expressions of CEBPα, PPARγ, and HO-1 at the translational level and obtained similar data to that from qPCR. BBR reduced the levels of CEBPα and PPARγ and increased HO-1 expression, whereas antagonism of PPARδ abolished the effect of BBR (Figure 6C,D).

We also analyzed the expression of PPARδ and the potential downstream targets in mice EWAT samples. Both the immunoblot and immunostaining results showed that the PPARδ level was dose-dependently increased by BBR, whereas the presence of GSK weakened this action (Figure 7A–C). The immunoblots of CEBPα, PPARγ, and HO-1 showed similar trends to those of the in vitro data (Figure 7A,B).

Therefore, we concluded that BBR activated PPARδ to regulate genes such as *Cebpα*, *Pparγ*, and *Ho-1*.

### 2.5. BBR Regulated Cebpα, Pparγ, and Ho-1 through Mediating the Transcriptional Function PPARδ

We asked how BBR mediates PPARδ to regulate downstream genes. We conducted a luciferase reporter assay, ChIP-PCR/qPCR, and site-directed mutagenesis to address this question. First, the results of the luciferase assay showed that the addition of BBR was able to regulate the activities of *Pparδ*, *Cebpα*, *Pparγ*, and *Ho-1* promoters (Figure 8A). ChIP-PCR helped to characterize the potential PPAR response element (PPRE) on each promoter, as shown in Appendix A. Next, the data from ChIP-qPCR with the potential PPREs, as identified in Appendix A, showed that BBR increased the promoters of *Pparδ* and *Ho-1* to bind PPARδ but reduced the promoter activities of *Cebpα* and *Pparγ* (Figure 8B). When the potential PPRE mutated, we found that the mutation reduced the ability of BBR to mediate the promoter activities (Figure 8B,C).

## 3. Discussion

The increasing trend in obesity, which affects a large proportion of the world’s population, has placed considerable pressure on public health globally. However, the current therapies are not as efficacious as anticipated. Though having some beneficial effects, these therapies are usually accompanied by adverse effects or risks. Therefore, novel, effective, and safe anti-obesity agents need to be developed.

BBR shows promise as an anti-obesity medication. BBR has been found to decrease appetite via downregulating the level of central neuropeptide Y, an orexigenic peptide, to stimulate food intake [22], or activating bitter-taste receptor signaling pathway [32] to inhibit mice weight gain. Targeting BAT also contributed to the anti-obesity activity of BBR, with a mechanism involved with the AMP-activated protein kinase–peroxisome proliferator-activated receptor-gamma coactivator alpha–silent-mating-type information regulation 2 homolog 1 pathway [23,27]. BBR has also been shown to decrease PPARγ and GAL-3 expression and increase GATA2, GATA3, and aP2 expressions to lessen adipogenesis [24,28,29]. Moreover, the modulation of the gut microbiome is another aspect of the mechanism of action of BBR against obesity, which includes the reduction in microbial diversity, the enrichment of bacteria producing short-chain fatty acids, and the restoration of the gut barrier [21,33,34].

Here, on the basis of PPARs functions, we are the first to report that BBR activates PPARδ to regulate downstream genes to inhibit adipogenesis, thereby suppressing obesity (Figure 9). Additionally, we are the first to find that the mediation of the downstream target genes is achieved by the BBR-provoked transcriptional function of PPARδ. As PPARs are vital regulators for a variety of metabolic diseases, novel anti-obesity agents targeting PPARs should be developed [16,35]. From our data, we found that PPARδ, rather than PPARα or PPARγ, plays an active role in the anti-obesity effect of BBR. However, only a few studies have discussed the effect of BBR on PPARδ, and none have revealed the underlying interaction mechanism. Zhou et al. reported that BBR increased PPARδ expression in diabetic rat livers to improve the metabolic disease state [36] and promoted PPARs and elongation factor b expression to produce the hypoglycemic and hypolipidemic effects [37].

From a totally different aspect, we explored the role of PPARs in the anti-obesity effect of BBR. Firstly, we applied gene silencing to determine which subtype of PPARs is the most important in the beneficial effect of BBR. Together with the validation of a specific inhibitor (GSK), we found that PPARδ strongly contributes to the efficacy of BBR against 3T3L1 preadipocytes and HFD-fed obese mice.

Regarding the consequence after the targeting of PPARδ by BBR, as PPARδ is a ligand-inducible transcriptional regulator and our previous study demonstrated that BBR is a potent ligand (Kd = 290 ± 92 nM) anchored inside PPARδ and interacts with N191, T25, and N306 residues [26,38], we suspected that PPARδ was activated by BBR in 3T3L1 cells and obese mice. We assayed the mRNA levels of endogenous PPARδ target genes, Ucp2 and Pdk4, to check whether they were activated, following the method of Dickey et al.’s work [39]. Both the in vitro and in vivo data supported the BBR-induced activation of PPARδ. Then, we screened which gene(s) were involved after PPARδ activation.

As transcriptional factors govern the transcription of target genes, qPCR analysis was a suitable tool for identifying the downstream genes of PPARδ [40]. Therefore, we performed qPCR analysis on 3T3L1 cells with BBR and GSK treatment to identify the downstream genes. Regarding some lipogenic factors such as acetyl-CoA carboxylase (ACC) and sterol regulatory-element-binding protein 1c (SREBP1c), BBR has been reported to affect their phosphorylation in 3T3L1 cells [41,42,43], which might not be regulated by a transcriptional factor. Here, we found that not all genes described in the literature were regulated by BBR treatment; for example, we did not obtain positive data for *Gal-3* and *aP2*. Although the other genes were regulated by BBR, only the activities of *Cebpα*, *Pparγ*, and *Ho-1* promoters were affected when PPARδ was antagonized by GSK. Therefore, we concluded that these three genes are the downstream targets after BBR activates PPARδ. Furthermore, we explored how activated PPARδ can regulate these downstream targets, which was achieved through transcriptional mediation on the promoters of the downstream genes. Plus, we determined the binding regions (PPRE) using a ChIP-PCR/qPCR assay and site-directed mutagenesis.

Therefore, we concluded that BBR follows a PPARδ-dependent mechanism to reduce lipid accumulation.

## 4. Materials and Methods

### 4.1. Chemicals and Reagents

BBR (purity > 95%, 10006427) was purchased from Cayman Chemical (Ann Arbor, MI, USA). GSK0660 (GSK) was bought from Santa Cruz Biotechnology (Dallas, TX, USA). Dulbecco’s modified Eagle medium (DMEM), fetal bovine serum (FBS), penicillin–streptomycin (PS), trypsin-EDTA, 3-(4,5-dimethylthPSiazol-2-yl)-2,5-diphenyltetrazolium bromide (MTT), and Lipofectamine™ 2000 transfection reagent were obtained from Thermo Fisher Scientific (Waltham, MA, USA).

Antibodies were obtained for protein detection: CEBPα (8178T, Cell Signaling, Danvers, MA, USA), PPARδ antibody (GTX113250, GeneTex, Irvine, CA, USA), PPARα (GTX101098, GeneTex), PPARγ (2435S, Cell Signaling), HO-1 antibody (GeneTex, GTX101147), β-Actin (AM4302, Thermo Fisher, Waltham, MA, USA), goat anti-rabbit IgG (H + L)-HRP conjugate (98164S, Cell Signaling), and goat anti-mouse IgG (H + L)-HRP conjugate (91196S, Cell Signaling).

Biochemical kits were obtained from Fisher Scientific (Waltham, MA, USA) and Beijing Solarbio Science and Technology (Beijing, China): Stanbio™ Triglyceride Liquid Reagent (2100430, Fisher Scientific), Stanbio™ Cholesterol Liquid Reagent (1010430, Fisher Scientific), Stanbio™ Glucose Liquid Reagent (1070125, Fisher Scientific), High-Density Lipoprotein Cholesterol (HDL) content assay kit (BC5320, Solarbio), Mouse TNFα ELISA kit (SEKM-0034, Solarbio), and MDA content assay kit (BC0025, Solarbio).

### 4.2. Cell Culture and Treatment

3T3L1 preadipocytes (ATCC, Manassas, VA, USA) were cultured in DMEM containing 10% FBS and 0.5% PS. BBR and GSK stock solutions were prepared using dimethyl sulfoxide and diluted to the designated concentration prior to cell treatment.

For the cell viability assay, various concentrations of BBR or GSK were added to 3T3L1 cells one day after cell seeding, and an MTT assay was performed after 48 h of treatment.

3T3L1 differentiation was induced with MDI [44]. The differentiation protocol is schematized in Figure 1B. BBR and/or GSK were added to treat cells on days 0, 2, or 4. Then, oil red O staining was performed on day 8 [44].

Gene silencing of 3T3L1 cells was performed on day 1 with lipofectamine-2000-mediated siRNA transfection, which was followed by BBR treatment.

### 4.3. Mice Experiment

Male C57BL/6J mice (4 weeks old) were provided by the Laboratory Animal Services Center at the Chinese University of Hong Kong and fed an HFD to induce obesity, according to Liu et al.’s work [45]. Briefly, the HFD (26 kg) was prepared by mixing the following ingredients: corn starch, 4.23 kg; casein, 6.8 kg; sucrose, 2.38 kg; cellulose, 1.7 kg; AIN 76 mineral mix, 1.2 kg; lard, 8.33 kg; soybean oil, 0.85 kg; AIN vitamin mix, 0.34 kg; choline bitartrate, 0.068 g; and L-cystine, 0.1 kg.

For administration to the mice, BBR was suspended in 0.05% sodium carboxymethyl cellulose–normal saline, and GSK was diluted in normal saline with 5% dimethyl sulfoxide. Mouse weight was recorded every week, and the plasma collected from the tail vein was used to detect triglyceride, total cholesterol, and glucose levels. Significantly higher mouse weight and detected parameters compared with those of the ND-fed mice were considered the successful establishment of the obese mouse model. Then, the obese mice were randomly divided into several groups (n = 6 per group); the 3-week drug treatment information is provided in Table 1. We continued to feed the mice the HFD during drug treatment. Mice weights were recorded every week. Plasma was collected from the tail vein to detect triglyceride, total cholesterol, glucose, HDL, TNFα, and MDA levels after BBR treatment.

### 4.4. Histological Examination and Immunostaining

Fresh samples (fat tissue and liver) were collected and fixed in 10% formalin solution for 24 h at room temperature. Samples were then embedded in paraffin, then sectioned into 5 or 10 μm thick sections for H&E staining or immunofluorescent staining. For immunofluorescent staining, the antigens were retrieved following an Abcam heat-induced epitope retrieval protocol. Then, the retrieved samples were used for PPARδ and DAPI staining.

### 4.5. qPCR and WB Analysis

The mRNA levels of *Pparα*, *Pparγ*, *Pparδ*, *Pdk4*, *Ucp2*, *Cebpα*, *Cebpβ*, *Ho-1*, *Gata2*, *Gata3*, *Gal-3*, *aP2*, and *β-Actin* were analyzed using Luna^®^ Universal qPCR Master Mix (M3003, New England Biolabs, Ipswich, MA, USA). The total protein was extracted with RIPA buffer (89900, Thermo Fisher) and quantified with a Pierce™ BCA Protein Assay Kit (23225, Thermo Fisher). Both qPCR and WB were performed as previously described [46]. Primers are listed in Appendix A.

### 4.6. Luciferase Reporter Assay and Chromatin Immunoprecipitation Coupled PCR/qPCR Assay

The interactions among PPARδ and promoters were studied via a luciferase reporter assay. The promoters of *Pparδ*, *Pparγ*, *Cebpα*, and *Ho1* were amplified from 3T3L1 cell gDNA using a Phusion^®^ High-Fidelity PCR Kit (New England Biolabs, Ipswich, MA, USA). Then, a luciferase reporter assay was performed as per our previous publication [26].

Next, we performed ChIP-PCR and ChIP-qPCR assays to elucidate the potential binding sites on the promoters [26]. The PPAR response element (PPRE) of each promoter was predicted online at http://www.classicrus.com/PPRE/ (accessed on 23 November 2020) (Appendix A), and the primers were designed accordingly (Appendix A). To confirm the PPREs, site-directed mutagenesis was used to mutate the potential regions for the luciferase reporter assay.

The primers used for promoter amplification and mutation are listed in Appendix A.

### 4.7. Statistical Analysis

All the experimental data were analyzed using one- or two-way ANOVA, followed by Bonferroni’s test using GraphPad Prism Version 6.0C (GraphPad Software Version 9.4.1 (458), USA). Data are expressed as means ± SEM. *p*-values less than 0.05 were considered statistically significant.

## 5. Conclusions

Here, our data showed that PPARδ is needed for BBR to exert an inhibitory effect on lipid accumulation to reduce obesity, which boosts our understating of the role of PPARδ in the effect of BBR and provides novel insight for developing BBR into an anti-obesity medication.

## Figures and Tables

**Figure 1 ijms-24-11600-f001:**
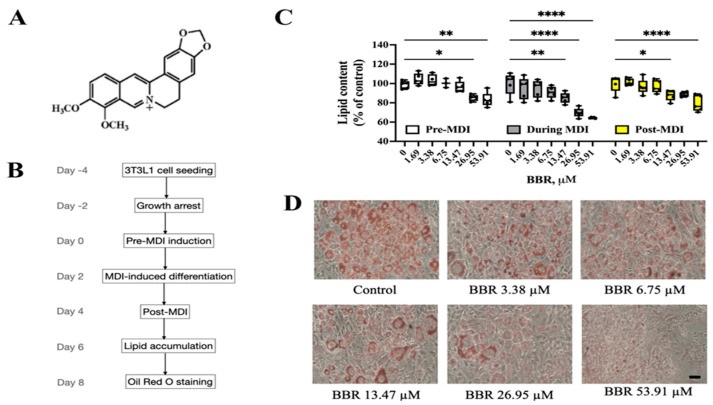
BBR decreased lipid accumulation in 3T3L1 cells. (**A**) Structure of BBR. (**B**) 3T3L1 cell differentiation protocol. (**C**) Lipid content in 3T3L11 cells with BBR treatment at different time points. (**D**) Representative images of oil red O staining in 3T3L1 cells with BBR treatment during MDI. Scale bar: 100 μm. The data represent the mean ± SEM (N = 3–4). **** *p* < 0.0001, ** *p* < 0.01, * *p* < 0.05.

**Figure 2 ijms-24-11600-f002:**
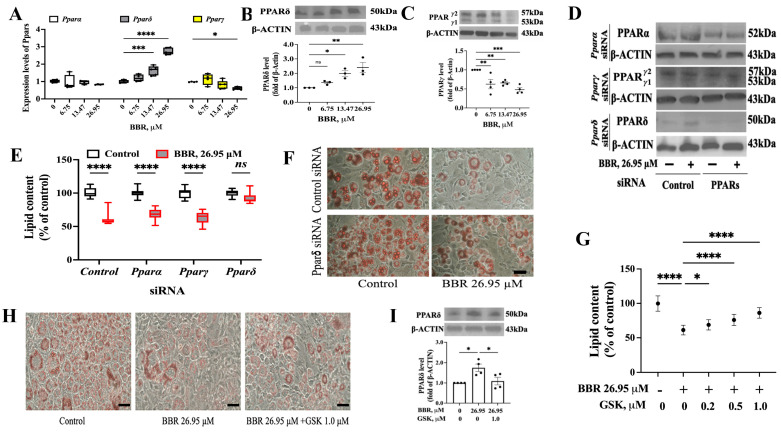
PPARδ was involved in the effect of BBR. (**A**) mRNA level of *Ppars* in 3T3L1 cells with BBR treatment for 48 h. (**B**,**C**) Immunoblot of PPARδ (**B**) and PPARγ1/2 (**C**) in 3T3L1 cells with BBR treatment. (**D**) Expression of PPARs with siRNA transfections. (**E**) Lipid content in siRNA-transfected 3T3L1 cells with BBR treatment. (**F**) Representative images of oil red O staining in control or *Pparδ* siRNA-transfected 3T3L1 cells. (**G**) Lipid content in 3T3L1 cells with BBR and GSK treatment. (**H**) Representative images of oil red O staining 3T3L1 cells with BBR and GSK treatment. (**I**) Immunoblot of PPARδ in 3T3L1 cells with BBR and GSK treatment. Scale bar: 100 μm. The data represent the mean ± SEM (N = 3–4). Compared with the control: **** *p* < 0.0001, *** *p* < 0.001, ** *p* < 0.01, * *p* < 0.05. ns: no significance.

**Figure 3 ijms-24-11600-f003:**
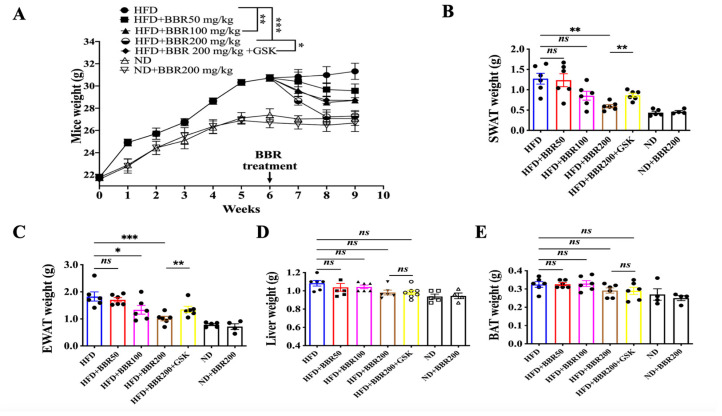
PPARδ-dependent effect of BBR on HFD mice weight, fat, and liver mass. (**A**) Mice weight with BBR treatment. (**B**–**D**) SWAT (**B**), EWAT (**C**), liver (**D**), and BAT mass (**E**). The data represent the mean ± SEM (N = 6). *** *p* < 0.001, ** *p* < 0.01, * *p* < 0.05. ns: no significance.

**Figure 4 ijms-24-11600-f004:**
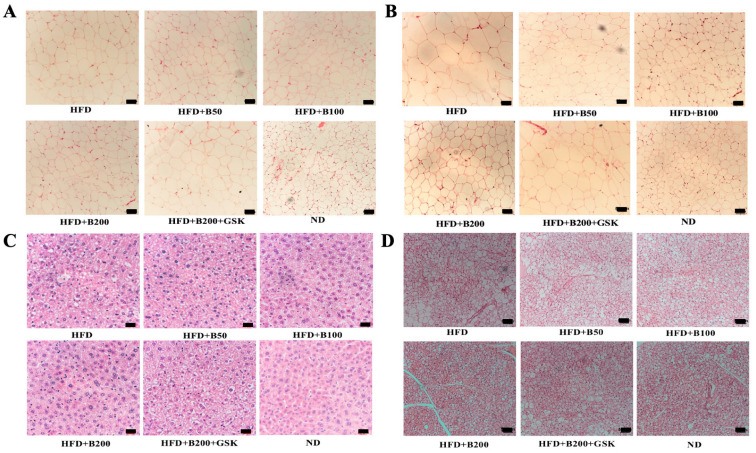
PPARδ-dependent effect of BBR on histological changes of fat and liver samples from HFD mice. (**A**–**D**) H&E staining of SWAT (**A**), EWAT (**B**), Liver (**C**), and BAT (**D**). Scale bar: 50 μm.

**Figure 5 ijms-24-11600-f005:**
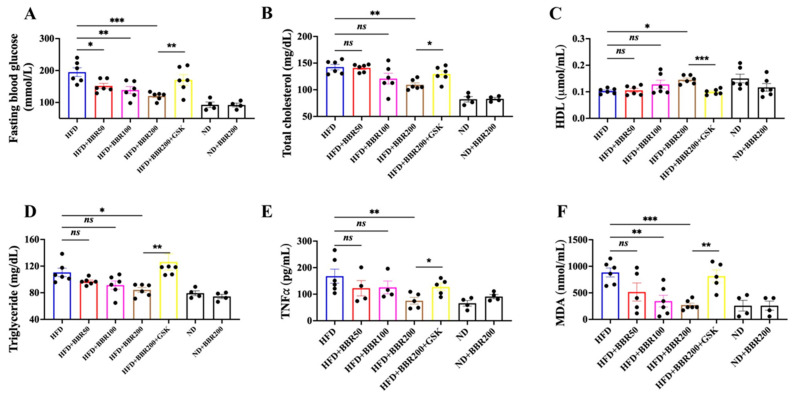
BBR followed a PPARδ-dependent manner to correct the disturbed biochemical plasma parameters. (**A**) Fasting blood glucose. (**B**) Total cholesterol. (**C**) HDL. (**D**) Triglyceride. (**E**) TNF*α*. (**F**) MDA. The data represent the mean ± SEM (N = 4–6). *** *p* < 0.001, ** *p* < 0.01, * *p* < 0.05. ns: no significance.

**Figure 6 ijms-24-11600-f006:**
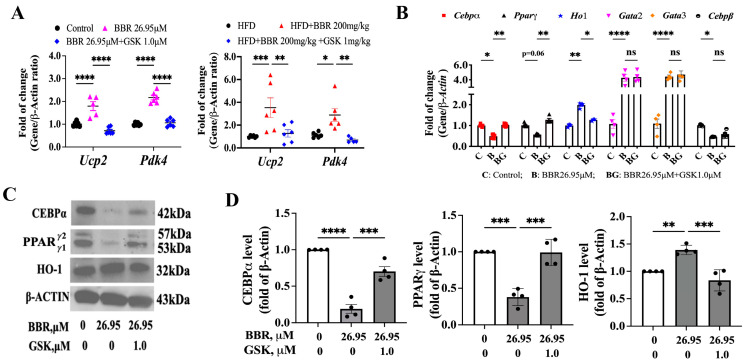
BBR activated PPARδ to mediate the downstream genes. (**A**) mRNA level of *Ucp2* and *Pdk4* in 3T3L1 cells (left) and HFD mice EWAT (right). (**B**) mRNA level of downstream genes in 3T3L1 cells with BBR and GSK treatment. (**C**,**D**) Immunoblots (**C**) and quantification data (**D**) of CEBPα, PPARγ1/2, and HO-1 in 3T3L1 cells with BBR and GSK treatment. The data represent the mean ± SEM (N = 3–4). **** *p* < 0.0001, *** *p* < 0.001, ** *p* < 0.01, * *p* < 0.05. ns: no significance.

**Figure 7 ijms-24-11600-f007:**
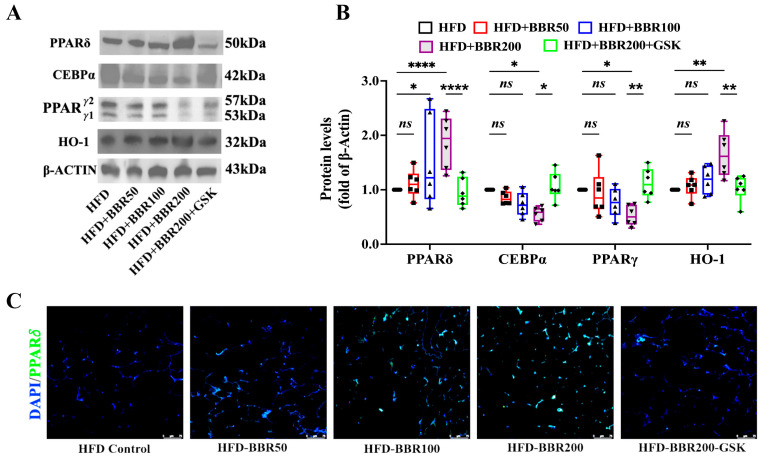
BBR regulated PPARδ, CEBPα, PPARγ, and HO-1 expression in HFD mice EWAT. (**A**) Immunoblots of PPARδ, CEBPα, PPARγ1/2 and HO-1. (**B**) Quantification data of PPARδ, CEBPα, PPARγ and HO-1. (**C**) Representative images of PPARδ staining. The data represent the mean ± SEM (N = 6). Scale bar: 75 μm. **** *p* < 0.0001, ** *p* < 0.01, * *p* < 0.05. ns: no significance.

**Figure 8 ijms-24-11600-f008:**
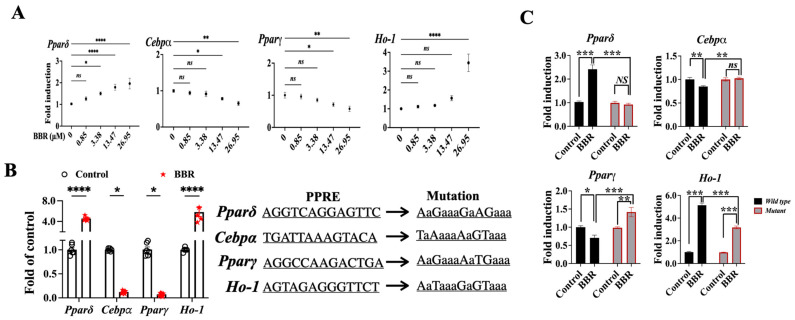
BBR activated PPARδ to trigger its transcriptional functions. (**A**) Luciferase assay of target gene promoter activities with BBR treatment. (**B**) ChIP-qPCR analysis of potential binding regions in downstream gene promoters. (**C**) Luciferase assay of the binding site (wild type and mutation) in downstream gene promoters with BBR treatment. The data represent the mean ± SEM (N = 4). **** *p* < 0.0001, *** *p* < 0.001, ** *p* < 0.01, * *p* < 0.05, ns: no significance.

**Figure 9 ijms-24-11600-f009:**
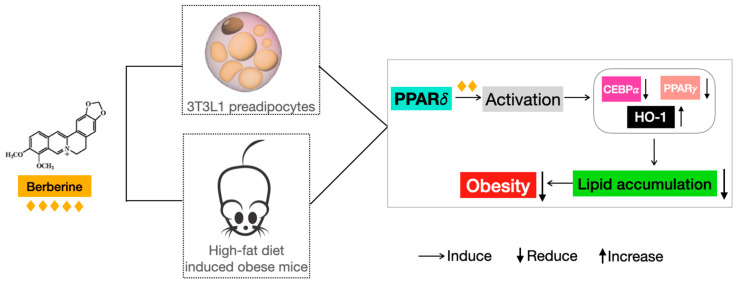
A graphic summary of BBR on obesity.

**Table 1 ijms-24-11600-t001:** Mice grouping information.

Groups	Treatment Information
ND	Normal C57BL/6J mice + normal diet+ vehicle
ND + BBR200	Normal C57BL/6J mice + BBR 200 mg/kg/day, i.g.
HFD	Obese C57BL/6J mice + HFD + vehicle
HFD + BBR50	Obese C57BL/6J mice + HFD + BBR 50 mg/kg/day, i.g.
HFD + BBR100	Obese C57BL/6J mice + HFD + BBR 100 mg/kg/day, i.g.
HFD + BBR200	Obese C57BL/6J mice + HFD + BBR 200 mg/kg/day, i.g.
HFD + BBR20 + GSK	Obese C57BL/6J mice + HFD + BBR 200 mg/kg/day, i.g. + GSK 1 mg/kg/day, i.p.

## Data Availability

Data are available upon request.

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
