# Peer review of "Berberine Reduces Lipid Accumulation in Obesity via Mediating Transcriptional Function of PPARδ"

_ijms, 2023, doi:10.3390/ijms241411600_

Round 1

Reviewer 1 Report (New Reviewer)

Congratulations for this good scientific work.

Author Response

Thanks for the comment.

Reviewer 2 Report (New Reviewer)

The manuscript is focused on the potential employment of berberine (BBR) in the reduction of lipid accumulation in obesity via mediating transcriptional function of peroxisome proliferator-activated receptors (PPARδ). Lipid accumulation, mouse weight, and biochemical parameters were determined to evaluate the efficacy of BBR. Additionally, gene silencing and specific antagonist treatment were used to verify the role of PPARs. The mechanism through which BBR affects PPARs was studied using qPCR, immunoblotting, immunostaining, and chromatin immunoprecipitation (ChIP)-PCR/qPCR analysis.

The paper is interesting since obesity is posing risks worldwide to people of all ages, genders and socioeconomic statuses. The experiments were carefully executed.

-English needs extensive revision. Too short sentences and redundant words appear in the manuscript e.g. Abstract, lines 12-14: Obesity is defined as a dampness-heat syndrome in traditional Chinese medicine, so heat-clearing herbs are used to treat obesity. Coptidis Rhizoma is an herb used to clear heat and eliminate dampness in obesity and its complications. Berberine (BBR), the main active compound in Coptidis Rhizoma, shows antiobesity effects.

Also, impersonal verbs are strongly encouraged.

Section 2.3. Why lines 130-132 are highlighted?

3.1. Chemicals and reagents. There no information on Coptidis Rhizoma and on the methodology used for BBR extraction.

5. Conclusions. They are too maigre. They should report the most striking results achieved in the study along with potential future applications.

Extensive revision is required (see comments)

Author Response

Comments and Suggestions for Authors

The manuscript is focused on the potential employment of berberine (BBR) in the reduction of lipid accumulation in obesity via mediating transcriptional function of peroxisome proliferator-activated receptors (PPARδ). Lipid accumulation, mouse weight, and biochemical parameters were determined to evaluate the efficacy of BBR. Additionally, gene silencing and specific antagonist treatment were used to verify the role of PPARs. The mechanism through which BBR affects PPARs was studied using qPCR, immunoblotting, immunostaining, and chromatin immunoprecipitation (ChIP)-PCR/qPCR analysis.

The paper is interesting since obesity is posing risks worldwide to people of all ages, genders and socioeconomic statuses. The experiments were carefully executed.

-English needs extensive revision. Too short sentences and redundant words appear in the manuscript e.g. Abstract, lines 12-14: Obesity is defined as a dampness-heat syndrome in traditional Chinese medicine, so heat-clearing herbs are used to treat obesity. Coptidis Rhizoma is an herb used to clear heat and eliminate dampness in obesity and its complications. Berberine (BBR), the main active compound in Coptidis Rhizoma, shows antiobesity effects.

Also, impersonal verbs are strongly encouraged.

Answer: We have revised the abstract and the language has been edited by MDPI. Please find the certificate as attached.

Section 2.3. Why lines 130-132 are highlighted?

Answer: We have removed the highlight.

3.1. Chemicals and reagents. There no information on Coptidis Rhizoma and on the methodology used for BBR extraction.

Answer: BBR (purity > 95%, 10006427) was purchased from Cayman Chemical (Ann Arbor, 292 Michigan, USA).

  1. Conclusions. They are too maigre. They should report the most striking results achieved in the study along with potential future applications.

Answer: We have polished the conclusion and thanks for the sugge

Reviewer 3 Report (New Reviewer)

Present study by Shou & Shaw et al. provides a comprehensive overview of a novel anti-obesity mechanism of Berberine regulated by PPAR-d receptor to bring reduction in lipid accumulation in 3T3L1 adipocytes and High fat diet induced mice using a variety of techniques. The major aim of the study is to elucidate the role of PPARs in Berberine treatment on obesity and the underlying molecular mechanisms. The question is interesting. However, there are some major concerns that needs to be addressed in the revised version as following:

Major comments

1.     It is not clear how the doses of Berberine were chosen?

2.     What are the two bands on PPAR-g in protein gels (Fig. 2c)? This needs to be mentioned in legend.

3.     PPAR-g levels do not seem to reduce linearly with increasing concentration of Berberine at translational levels as seen from the protein gels (Fig. 2c). Have authors tried out this experiment with all levels of Berberin concentrations (e.g – 0 to 134.7 uM)?

4.     What are the effects of increasing Berberine concentrations on PPAR-a levels at the translational level?

5.     It is hard to visualize successful gene silencing using western blot as authors stated and showed in Fig 2d unless images with PPARs levels before and after siRNA treatment at both transcriptional and translational levels are shown in the form of gels and compared quantitatively. It is highly recommended to add both the qualitative and quantitative data regarding siRNA efficiency.

6.     How do authors label lipid vacuoles inside the liver cells? It is hard to make clear conclusions regarding effects of increasing concentration of Berberine on reducing the lipid vacuoles inside the liver simply by visualizing H & E staining data (Figure 4C). It is recommended to quantitate this data as well and incorporate that in the revised manuscript.

7.     A study (https://www.ncbi.nlm.nih.gov/pmc/articles/PMC8375237/) by Xu et al. 2021 revealed Berberine as a selective PPAR-γ activator to promote adipose tissue remodeling and thermogenesis. However, present study found that Berberine rather increased PPAR-δ level, and decreased PPAR-? expression in a dose-dependent manner. Also, Berberine does not seem to function through PPAR-g to control lipid levels in transfected 3T3L1 cells in present study. Why there are differences in types of activated PPARs modulators induced by Berberine across different studies studying similar physiological questions?

Minor comments

1.     Figures 2G,2H and 2I are labelled incorrectly as 1G,1H and 1I in line numbers 125,126 respectively.

2.     Minor spelling mistakes found that can be corrected.

English/spell check scan throughout the manuscript required. Minor errors found.

Author Response

Major comments

  1. It is not clear how the doses of Berberine were chosen?

Answer: We treated 3T3L1 cells with  a series concentrations of berberine from 0 – 53.91 uM (0-20ug/mL) and chose the effective doses (13.47 – 53.91 uM) (Fig. 1C), which was in accordance with others’ publications (PMID: 27938388; 31527742; 26197511).

  1. What are the two bands on PPAR-g in protein gels (Fig. 2c)? This needs to be mentioned in legend.

Answer: There two isoforms, gamma1 and gamma2 ,encoded by a single PPAR gamma gene. We have added a label in Fig. 2c and a brief description in the legend. For PPAR gamma immunoblot quantification, we combined these two bands’ intensities following the publications (PMID: 36900533; 36333974).

  1. PPAR-g levels do not seem to reduce linearly with increasing concentration of Berberine at translational levels as seen from the protein gels (Fig. 2c). Have authors tried out this experiment with all levels of Berberin concentrations (e.g – 0 to 134.7 uM)?

Answer: We have re-performed the experiment and provided a new protein gel image in Fig. 2C. We did not detect the PPARg expression at different berberine concentrations. In Fig 1C, not all the dose of berberine showed the effects on 3T3L1 differentiation, thus we chose several effective doses (6.75, 13.47 and 26.95 uM) in the following experiments (Fig 2A and 2C).

  1. What are the effects of increasing Berberine concentrations on PPAR-a levels at the translational level?

Answer: The level of PPARa protein was not altered after treatment of increasing berberine doses (shown below), which was consistent was our qPCR data in Fig 2A.

  1. It is hard to visualize successful gene silencing using western blot as authors stated and showed in Fig 2d unless images with PPARs levels before and after siRNA treatment at both transcriptional and translational levels are shown in the form of gels and compared quantitatively. It is highly recommended to add both the qualitative and quantitative data regarding siRNA efficiency.

Answer: Thanks for the suggestion. The quantitative data has been added in Fig. S1C.

  1. How do authors label lipid vacuoles inside the liver cells? It is hard to make clear conclusions regarding effects of increasing concentration of Berberine on reducing the lipid vacuoles inside the liver simply by visualizing H & E staining data (Figure 4C). It is recommended to quantitate this data as well and incorporate that in the revised manuscript.

Answer: Thanks. We have calculated the percentage of lipid vacuoles area to the total area with Image J and the data are shown in Fig. S3C.

  1. A study (https://www.ncbi.nlm.nih.gov/pmc/articles/PMC8375237/) by Xu et al. 2021 revealed Berberine as a selective PPAR-γ activator to promote adipose tissue remodeling and thermogenesis. However, present study found that Berberine rather increased PPAR-δ level, and decreased PPAR-? expression in a dose-dependent manner. Also, Berberine does not seem to function through PPAR-g to control lipid levels in transfected 3T3L1 cells in present study. Why there are differences in types of activated PPARs modulators induced by Berberine across different studies studying similar physiological questions?

Answer: PPAR-? is one of the master genes that regulate 3T3L1 differentiation. Both others’ and our data showed that berberine treatment reduced PPAR-? level (Fig 2A, 2C, 7A and 7B; PMID: 16890192, 31527742, 25928058 etc.) to reduce 3T3L1 cell differentiation. Moreover, our data suggested that berberine regulated PPARd’s transcriptional function to downregulate the PPAR? expression to reduce lipid accumulation, which renders a novel mechanism to elucidate the role of PPARs in the effect of berberine on 3T3L1 cells. However, the article you mentioned aimed at the effect of berberine on adipose tissue remodeling and thermogenesis, which is another mechanistic aspect of BBR and different from our findings.

The reason why berberine showed different action on PPARs in similar physiological questions remains unclear. Besides our data, there are many other examples. For instance, in streptozotocin-induced diabetic rats, Zhou et al reported that BBR treatment reduced PPAR? expression to improve glucolipid metabolism (PMID: 18520050), however, Mahmoud et al reported that berberine upregulated PPAR? to attenuate hyperglycaemia and its associated oxidative stress and inflammation (DOI: 10.7324/JAPS.2017.70401). Though it is of great importance to explain why berberine shows different actions on PPARs in similar diseases, our manuscript did not aim to answer this question.

Minor comments

  1. Figures 2G,2H and 2I are labelled incorrectly as 1G,1H and 1I in line numbers 125,126 respectively.

Answer: Thanks. We have corrected the errors in the revised manuscript.

  1. Minor spelling mistakes found that can be corrected.

Answer: We have corrected the mistakes.

Round 2

Reviewer 2 Report (New Reviewer)

The authors have satisfactorily addressed all remarks.

Author Response

Thanks for the comment.

Reviewer 3 Report (New Reviewer)

Authors have significantly improved the manuscript as recommended. However, I cannot see the quantitative representation of Fig 2d in supplementary that was asked in point number 5.

Author Response

We have added the quantitative data of PPARs level with siRNA treatment has been added in Fig. S1C (supplementary figures).

This manuscript is a resubmission of an earlier submission. The following is a list of the peer review reports and author responses from that submission.

Round 1

Reviewer 1 Report

General comments

It is a quiet interesting study. I believe that this manuscript tries to investigate the effect of bebeerine in 3T3-L1 adipogenesis (in vitro) and in obese mice (in vivo) model. However, there are also some of weaknesses. The following are my comments and critique.

Major

1. Have you checked C/EBP beta expression (protein/RNA) level during MDI with or without BBR? It is very important to check C/EBP beta level because it is a master regulator of adipogenesis.

2. For mice study, how many times repeated? It looks just once. If so, please repeat at least twice.

3. Figure 7B, please perform ChIP-qPCR instead of ChIP-PCR. Please include which PPREs in each gene were used?

4. Why did GSK0660 reduce PPARdelta protein level? I believe antagonist usually regulates the activity of target, not decrease protein level. Please explain.

Minors

1.      How about PPAR delta expression during adipogenesis (from Day 0 to Day 8)?

2.      Lack of detailed information of figures and legends. Please update.

3.      Have you checked about the expression of genes related to FAO, lipogenesis or thermogenesis?

Moderate editing of English language required

Author Response

  1. Have you checked C/EBP beta expression (protein/RNA) level during MDI with or without BBR? It is very important to check C/EBP beta level because it is a master regulator of adipogenesis.

Response: Thanks for the advice. We added the qPCR data of C/EBP beta in the revised Fig 6B, though BBR decreased the mRNA level of C/EBP beta, GSK (PPARd antagonist) treatment did not reverse this action. So we did not explore it further.

  1. For mice study, how many times repeated? It looks just once. If so, please repeat at least twice.

Response: Actually, we have repeated the mice experiment for once (N=4), but we did not show the repeat data in the manuscript. Please find the repeat data as follow:

As the Three Rs (3Rs, Replacement, Reduction and Refinement) are guiding principles for ethical use of animals in research, we were approved to use at most 10 mice per group by the anima ethic committee in the Chinese University of Hong Kong. Therefore, we firstly performed a pilot study (the data presented here) to check whether the experimental settings, like the diet, drug doses or treatment period, were reasonable. Then, we performed the formal study (the data presented in the manuscript, N=6). Both these two batches of data showed the similar effect of BBR. Besides, three doses of BBR were given to mice and there showed the dose-dependent anti-obesity effect of BBR. Therefore, we believe the data we have are accurate.

  1. Figure 7B, please perform ChIP-qPCR instead of ChIP-PCR. Please include which PPREs in each gene were used?

Response: Thanks for the advice. ChIP-PCR helped us identify which PPRE was responsible for the binding affinity. The qPCR data has been added in revised Fig.8B. 

  1. Why did GSK0660 reduce PPARdelta protein level? I believe antagonist usually regulates the activity of target, not decrease protein level. Please explain.

Response: The effect of GSK0660 to reduce PPARd protein level was also found in publications like PMID: 30622619, PMID: 27789312, PMID: 37226204, PMID: 28672855, PMID: 30584345 etc. Our previous data have found that berberine could trigger Ppard promoter to bind PPARd in stroke and liver cancer, which indicates the mechanism of berberine to promote PPARd protein level (PMID: 37148713, PMID: 35093536). Here, we observed a similar phenomenon in obesity. This self-regulative effect has also been reported in PPARa after ligand activation (PMID: 28821620). We believe most antagonists regulate the activity of target, but there might be some exceptional cases in transcriptional factors. Besides PPARs, antagonists reduced protein levels have been reported in C/EBPβ (PMID: 36121385) and c-Myc (PMID: 35414060).

Minors

  1. How about PPAR delta expression during adipogenesis (from Day 0 to Day 8)?

Response: We have monitored the PPAR delta expression during adipogenesis and found that PPAR delta expression remained unaltered without BBR treatment. But if BBR was added on day 2, its levels were significantly promoted (see the figure below).

However, in this study, we focused on which timepoint of BBR addition showed the most potent inhibition effect on 3T3L1 preadipocyte differentiation. Moreover, BBR was added to incubate with the cells for 48h, rather than the whole differentiation period, so we did not include the data in the manuscript.

  1. Lack of detailed information of figures and legends. Please update.

Response: We have added detailed information in the revised manuscript.

  1. Have you checked about the expression of genes related to FAO, lipogenesis or thermogenesis?

Response: Actually, we did not focus on those mentioned genes. Here, we aimed at finding out which PPAR was pivotal to BBR-related suppressive effect on 3T3L1 cells and high fat diet induced obese mice. As we found that BBR reduced lipid accumulation, FAO and thermogenesis were not taken into consideration. For lipogenesis genes, like PPARg, ACC, Fabp4 (also known as Ap2), Srebp-1c, we analyzed PPARg and Fabp4 levels in the manuscript. For ACC and Srebp-1c, they need to be phosphorated before executing their function in lipogenesis. Thus, they might not be directly regulated by a transcriptional factor.

Reviewer 2 Report

The data obtained in the article are of undoubted scientific interest, but there are a huge number of errors that need to be corrected before publication.

In the introduction there are statistics for 2016, why more recent data didn’t use?

Why should the introduction focus on Covid-19?

It would be better to change the names of drugs to international non-proprietary names.

Line 63. How about "no"? What about saroglitazar in India?

Line 73. "has been reported show" - something missing here

The section names in the Results represent conclusions, this should not be the case.

Line 87. What does "non-toxic doses" mean, values need to be stated and justification for non-toxic.

The authors seem to confuse adipogenesis and cellular fat accumulation, it should be clarified what the authors mean, as adipogenesis is the growth of new adipocytes, which has not been assessed in in vitro tests.

Fig. 1 why such uneven concentrations of berberine were used, why not whole numbers?

Fig. 2 what is GSK, need to give the abbreviations. Line 116 - WB?

Lines 132-134 - should be moved to materials and methods

Fig. 3 Very small images especially B-F, impossible to analyze.

Description of animal methodology needs significant correction. Not specified the origin of the mice, need to add details of HFD rather than just a reference to the work, clarify which parameters were considered to be obesity parameters, not specified how the mice were bled to determine biochemical parameters before berberine was administered. It is not specified how blood plasma was obtained, how berberine was dissolved, were the ND group mice injected with a solvent? Again, ND and NB are not transcribed.

Table 1. Double "mice".

Author Response

The data obtained in the article are of undoubted scientific interest, but there are a huge number of errors that need to be corrected before publication.

In the introduction there are statistics for 2016, why more recent data didn’t use?

Response: We have updated the data in the revised manuscript.

Why should the introduction focus on Covid-19?

Response: COVID-19 pandemic worsened the obesity and the obeses are easy to develop severe Covid-19 pneumonia. Here, we just point out the increased obesity trend and negative outcome.

It would be better to change the names of drugs to international non-proprietary names.

Response: We have replaced all the drug names.

Line 63. How about "no"? What about saroglitazar in India?

Response: Saroglitazar (Lipaglyn™) is a dual PPAR-α and γ agonist that was approved for use by the Drug Controller General of India for the treatment of diabetic dyslipidemia and hypertriglyceridemia that is not controlled by statin therapy. Though it might do some good to obesity, the official indications of Saroglitazar did not include obesity. Therefore, we claimed that no commercial PPAR-dependent anti-obesity agents have been developed.

Line 73. "has been reported show" - something missing here

Response: Thanks. A “to” has been added.

The section names in the Results represent conclusions, this should not be the case.

Response: We have revised the names.

Line 87. What does "non-toxic doses" mean, values need to be stated and justification for non-toxic.

Response: Thanks. Non-toxic doses here mean the dose will not reduce cell viability. High doses (toxic) might induce cell death, and in this case the anti-differentiation effect of BBR will not be evaluated accurately.

The authors seem to confuse adipogenesis and cellular fat accumulation, it should be clarified what the authors mean, as adipogenesis is the growth of new adipocytes, which has not been assessed in in vitro tests.

Response: Thanks for the valuable comment. We aimed at cellular fat accumulation and have replaced “adipogenesis” with “cellular fat accumulation” in the revised manuscript.

Fig. 1 why such uneven concentrations of berberine were used, why not whole numbers?

Response: We have weighed berberine (mg with whole numbers) and converted the dose from μg/ml to μM, for example, 20 μg/ml corresponds to 53.91 μM.

Fig. 2 what is GSK, need to give the abbreviations. Line 116 - WB?

Response: Thanks. GSK is short for GSK0660, and WB is short for western blot.

Lines 132-134 - should be moved to materials and methods

Response: This part has been revised.

Fig. 3 Very small images especially B-F, impossible to analyze.

Response: We have split the original Fig 3 into revised Fig 3 and Fig 4. The concerned images have been enlarged.

Description of animal methodology needs significant correction. Not specified the origin of the mice, need to add details of HFD rather than just a reference to the work, clarify which parameters were considered to be obesity parameters, not specified how the mice were bled to determine biochemical parameters before berberine was administered. It is not specified how blood plasma was obtained, how berberine was dissolved, were the ND group mice injected with a solvent? Again, ND and NB are not transcribed.

Response: Thanks. We have added the concerned information in the revised manuscript.

For the obesity development, we monitored mice weight and detected the plasma levels of total cholesterol, glucose and triglycerides. We found that 6-week of HFD administration significantly increase these mentioned parameters, therefore indicating the obese state. We have mentioned this part in 2.3 “The in-vivo anti-obesity effect of BBR depended on PPARδ”.

Plasma collection method, and vehicle for berberine solution have been added.

Table 1. Double "mice".

Response: Deleted.

Round 2

Reviewer 2 Report

I still believe that header titles in the results section should be changed to more common ones and should not be as conclusion. Other my remarks were corrected by the authors. 

Author Response

Thanks for the comment. We have revised the header titles, please refer to the revised manuscript.
